# On the Influence of Aging on Classification Performance in the Visual EEG Oddball Paradigm Using Statistical and Temporal Features

**DOI:** 10.3390/life13020391

**Published:** 2023-01-31

**Authors:** Nina Omejc, Manca Peskar, Aleksandar Miladinović, Voyko Kavcic, Sašo Džeroski, Uros Marusic

**Affiliations:** 1Department of Knowledge Technologies, Jožef Stefan Institute, 1000 Ljubljana, Slovenia; 2Jožef Stefan International Postgraduate School, 1000 Ljubljana, Slovenia; 3Institute for Kinesiology Research, Science and Research Centre Koper, 6000 Koper, Slovenia; 4Biological Psychology and Neuroergonomics, Department of Psychology and Ergonomics, Faculty V: Mechanical Engineering and Transport Systems, Technische Universität Berlin, 10623 Berlin, Germany; 5Department of Ophthalmology, Institute for Maternal and Child Health-IRCCS Burlo Garofolo, 34137 Trieste, Italy; 6Institute of Gerontology, Wayne State University, Detroit, MI 48202, USA; 7International Institute of Applied Gerontology, 1000 Ljubljana, Slovenia; 8Department of Health Sciences, Alma Mater Europaea—ECM, 2000 Maribor, Slovenia

**Keywords:** aging, EEG, machine learning, classification, BCI, visual oddball study

## Abstract

The utilization of a non-invasive electroencephalogram (EEG) as an input sensor is a common approach in the field of the brain–computer interfaces (BCI). However, the collected EEG data pose many challenges, one of which may be the age-related variability of event-related potentials (ERPs), which are often used as primary EEG BCI signal features. To assess the potential effects of aging, a sample of 27 young and 43 older healthy individuals participated in a visual oddball study, in which they passively viewed frequent stimuli among randomly occurring rare stimuli while being recorded with a 32-channel EEG set. Two types of EEG datasets were created to train the classifiers, one consisting of amplitude and spectral features in time and another with extracted time-independent statistical ERP features. Among the nine classifiers tested, linear classifiers performed best. Furthermore, we show that classification performance differs between dataset types. When temporal features were used, maximum individuals’ performance scores were higher, had lower variance, and were less affected overall by within-class differences such as age. Finally, we found that the effect of aging on classification performance depends on the classifier and its internal feature ranking. Accordingly, performance will differ if the model favors features with large within-class differences. With this in mind, care must be taken in feature extraction and selection to find the correct features and consequently avoid potential age-related performance degradation in practice.

## 1. Introduction

The loss of communication pathways that enable interaction with the environment often has serious consequences for the patient. With ongoing advances and increasing acceptance of robotics and brain-computer interfaces (BCI) in neurorehabilitation, some negative impacts can be mitigated. When it is not otherwise possible, various devices (e.g., a neuroprosthesis, a wheelchair, a speller, or a computer cursor, see [1]) can be controlled by an electroencephalogram (EEG). EEG has many suitable properties—it is a non-invasive, safe, relatively simple, and inexpensive neuroimaging technique with a high temporal resolution that provides extensive information about brain activity. On the other hand, the gathered EEG data are complex, non-linear, non-stationary, high-dimensional signals that are contaminated with noise and inter-individual differences [2,3,4]. Consequently, efficient and reliable EEG data analysis is essential [5].

EEG BCIs provide the user with an alternative way of acting on the world. While measuring EEG activity, important features are extracted from the brain signal and analyzed. Then, the obtained information is in real-time translated into specific command, executed by a device that is implementing the user’s intentions [6]. Among the most relevant and common EEG features used in BCIs are the event-related potentials (ERPs) [5,7,8]. ERPs are relatively small voltage spikes generated by the underlying cascaded cortical processes and are triggered by external events such as visual stimuli. The most common visual ERPs are the positive P1 and the negative N1 waves, recorded from the occipital electrodes around 100 ms after the stimulus onset [9]. When the facial stimulus is presented, the N1 wave includes the N170 component that occurs at approximately 170 ms and is strongly related to the early perception of faces [10]. Beside early visual components, the P3 wave is an important late endogenous positive component, occurring around 300 ms after the stimulus presentation. The P3, most prominently observed in the parieto-central regions, is classically elicited in an oddball study, where randomly occurring rare stimuli capture attention in between the less relevant, frequent stimuli [11].

An important point we explore further in this study is that a healthy aging brain, ignoring individual variability, exhibits a number of differences in EEG activity compared to the brain of young adults [12,13,14]. While the basic neural mechanisms are maintained, older adults show an overall reduction in power across all frequency bands, indicative of general slowing [15,16,17]. Moreover, many studies have shown ERP age-related changes. Although ERPs are highly task-specific, the general consensus is that aging delays the latency of the visual (and other sensory) ERPs [12,18,19,20,21], while the effects on the sensory ERP amplitudes are not so clear. Studies mainly report no differences [19,20,21] or a decrease in amplitude [21,22,23,24]. Interestingly, following the compensatory hypothesis of aging [25], it has been shown that amplitudes of the posterior components decrease with age, while the amplitudes of the components in the prefrontal areas increase with age in order to maintain good performance [22]. As neurorehabilitation often includes older patients, possible brain changes must be considered when designing the EEG BCI rehabilitation process using automatic machine learning approaches.

To better understand how age-related differences might affect the BCI closed-loop rehabilitation process, we performed a binary classification task with a visual EEG oddball trial that elicited both early visual ERPs and the P3 wave. When designing a classification task, the classifier and feature selection are the two most important steps. From the plethora of possible classifiers, we decided to test the aging effect with (i) three linear classifiers, namely, Linear Discriminant Analysis (LDA), Support Vector Classifier (SVC) with no kernel, and Logistic Regression (LR), (ii) three nonlinear classifiers, namely Decision Tree Classifier (tree), K-Nearest Neighbours (KNN) and the SVC with radial basis function (RBF) kernel, (iii) as well as three ensemble methods, namely Random Forest (RF), Adaptive Boosting (AdaBoost) and Extreme Gradient Boosting (XGB). The classifiers were chosen as the representatives of each class, as they commonly appeared in the EEG BCI-related literature [1,6,8,26,27] or are classical representatives of specific method type. Especially LDA [28,29,30] and SVM [31,32,33] have been highly used in practice, but also KNN [34], LR [35] and RF [36]. Despite the current high interest in deep neural network (DNN) architectures (see e.g., [6,37,38,39,40,41,42]), these were not considered here due to the limited training data available and the number of hyperparameters that would need to be defined.

When using standard classifiers for high-dimensional EEG data, the feature extraction and selection steps are of great importance. A recent article reviewed 147 EEG features that can be extracted from the EEG signal [27], grouped by features in the time-domain, frequency-domain, time-frequency domain, nonlinear features, entropy, spatio-temporal features, and complex networks features. The review does not imply that certain features are necessarily better than others, but it does suggest that using various feature types can lead to improved classification performance. Following this notion, we created two types of datasets for training. The first was time-independent and contained relevant ERP statistical parameters, extracted from the time series. For instance, each ERP component can be parameterized by the peak amplitude, the mean amplitude, the peak latency, and the fractional 50% peak latency [43]. The second dataset included time-dependent features, particularly the amplitude and the signal power at a given frequency over time. While too large to be useful in an online application, such an approach enables better investigation of the possible age differences and variations in feature importance over time.

### Related Work

As already described, the age-related changes in ERPs properties have been known for a while and many studies have focused on using classical (e.g., [31,36]) and increasingly deep learning classifiers [39] to pick up these differences and determine age or other biometric traits from the EEG recordings. For example, to classify the age of participants based on the EEG features, researchers have used RF [36] and SVM-RBF [31], as well as DNNs [39]. The DNN method was a variant of long-short-term memory (LSTM) architecture and reached an age prediction accuracy of 93.7%. In the BCI applications, however, the goal is to reduce the performance dependency on biometric traits. Many studies have already looked at whether and how aging affects BCI performance. For example, it has been shown that the accuracy of the LDA classifier in vibro-tactile EEG BCI tasks was more than 15.9% lower for older subjects, as compared to that of the younger subjects [30]. Similarly, elderly people had slightly worse BCI performance on an eight-target EEG BCI spelling interface using code-modulated visual evoked potentials as features [44]. Another study tested the age-related drop in performance on the EEG BCIs with visual stimulations and also showed lower performance in the elderly when they used motion-onset VEP and steady-state-response VEP (SSMVEP & SSVEP) as features [45]. Lastly, Volosyak et al. [46] demonstrated a significant difference between the performance of younger and older subjects, again using the SSVEP features. The accuracy was 98.49% for young and 91.13% for older participants. While the presented studies clearly demonstrate the age-related differences, they only vaguely point to the possible reasons and solutions. Our goal was to dig deeper into the analysis of the classification results to obtain an explanation of what is a direct consequence of observed performance differences. Rather than focusing on an increase in predictive power, we tried to focus on the explainability of the results. With that in mind, we also utilized a great time resolution of the EEG and pointed to the possible caveats and solutions when the same classifier is used on people with different biometric traits, such as age.

## 2. Methods

The general workflow of the study is graphically depicted as a diagram in Figure 1. In this section, the general steps are further explained in detail.

### 2.1. Participants

Data (*n* = 79) for this analysis were collected in the framework of two different studies. Data from the first study included 46 older participants, 3 of whom were excluded during preprocessing stage due to excessive non-brain-related artifacts. Data from the second study comprised 33 young participants, 2 of whom were initially excluded due to nausea during EEG measurements, and 4 later due to non-brain-related artifacts. Thus, the final classification task included data from 43 older participants (27 females, mean age = 67.4 ± 5.5 years) and 27 young participants (14 females, mean age = 34.2 ± 2.3 years). The average number of years of education was 13 for the older participants and 16 for the younger participants. None had a history of psychiatric or neurological disorders, and all reported normal or corrected-to-normal vision. To exclude for mild cognitive impairment (MCI), the Montreal Cognitive Assessment (MoCA) screening tool was administered to the older group. All older participants scored above the threshold for MCI (>26/30 points), as was defined by Nasreddine et al. in their original paper [47]. All procedures were carried out in accordance with the ethical standards of the 1964 Declaration of Helsinki and were approved by the National Medical Ethics Committee (No. KME 57/06/17). Written informed consent was obtained from all participants prior to study enrollment.

### 2.2. Visual Oddball Task

A simple visual two-stimuli oddball task (see Figure 2) was used to obtain EEG activity of visual stimulus processing. Participants were seated in front of a 17-inch computer screen, at a 50 cm distance. On a black background, a white empty square (4.1 cm${}^{2}$) was displayed in the center. Inside the square, 150 ms long stimuli appeared throughout the task. The task involved the presentation of 124 (84%) frequent non-target stimuli (white solid square, 4.1 cm${}^{2}$) and 23 (16%) rare, randomly occurring, target stimuli (Einstein’s face, 4.1 cm${}^{2}$), with on average 669.6 ms interstimulus interval (SD = 11.8 ms). Participants were instructed to silently count the number of times the target stimulus occurred and report the sum at the end of the presentation. No motor response was required. While it has been shown that the ERP waveform to the visual oddball paradigm differs between overt (keypress) and covert (silent counting) conditions, the components differed only in their magnitude. Based on the topography and dipole modeling, the same brain areas become activated [48]. While we saved the behavioral results, which was the sum of the target trials at the end of the task, we have not used it in the analysis, as all participants correctly counted the number of target stimuli, except for one participant, who missed the correct sum by one.

### 2.3. EEG Analysis

EEG activity in the visual oddball task was recorded using a 32-channel EEG set consisting of 32 Ag/AgCl electrodes arranged according to the international 10–20 system. The signal was recorded at a frequency of 512 Hz and a resolution of 32 bits. The collected data were preprocessed using EEGLAB, an open source signal processing software [49] within the MATLAB program [50]. Recordings were visually inspected and high-noise sections were manually removed. Data were down-sampled to 256 Hz, filtered using 1 Hz high-pass and 40 Hz low-pass in-built EEGLAB filters and re-referenced to average. Using the Clean Rawdata plug-in, each channel was seemingly interpolated and correlated to its neighboring channels. All channels which had a correlation to their neighboring channels below 0.85 for more than half of the recording time, were then actually interpolated. Independent component (IC) analysis was used to obtain eye and muscle movement-specific information, as well as non-brain related information (e.g., line noise), which was then removed with the help of ICLabel classifier [51]. All ICs that were classified as the non-brain-related category above 85% certainty were removed from further analysis.

### 2.4. ERP Analysis

The continuous preprocessed data were then epoched into 0.9 s long bins that started 200 ms before the stimulus. Additionally, we removed epochs with an absolute peak amplitude over 100 µV. If more than 25% of all epochs were removed, the entire recording was excluded from the study. The minimum number of retained trials per participant was 129 (87.8%) trials, while the average was 145.9 trials (99.2%).

### 2.5. Extraction of Temporal Features

Forty features were calculated from the multi-dimensional 32×256×70×147 (channels, time points, participants, trials) data matrix. Four features corresponded to the voltage levels of four central spatial clusters at the channel level. The occipital cluster represented the average voltage fluctuations at O1, Oz and O2 channels, the parietal at P3, Pz and P4 channels, the central cluster C3, Cz and C4 channels and the frontal represented the average of F3, Fz and F4 channels. The remaining 36 features corresponded to the estimated spectral power (in dB) at 9 linear-spaced frequencies from 4 Hz to 36 Hz, measured at all four electrode clusters. The power was calculated based on the time-frequency analysis using the Fast Fourier spectral decomposition with a sliding Hann window of 32 time points. Analysis was performed using EEGLAB. To obtain power estimates of sufficient quality, 16 time points were removed from each end of the time series. The new reduced time interval between −136 ms and 734 ms (223 time points) was consequently used. Further mathematical details of the temporal feature construction can be seen in Appendix B. Additionally, we calculated mutual information between the features and a target and used it as a feature selection method to decrease the number of features to the best eight, as shown in Appendix D. The filter selection method was chosen because it is model agnostic and less prone to overfit on small datasets. In summary, the data in the classification task were composed of 10,214 trials (split between 70 participants) × 223 time points × 8 features. Selected features in the final dataset were occipital and central amplitudes, power at 4 Hz and 8 Hz in the occipital channels, power at 4 Hz, 8 Hz, and 12 Hz in the central channels, and power at 4 Hz in the frontal channel cluster. Figure 3 shows the trial-averaged features, grouped by age and stimulus type.

### 2.6. Extraction of Time-Independent Statistical ERP Features

The same multi-dimensional 32×256×70×147 (channels, time points, participants, trials) data matrix was used to create the second dataset. For every subject, the most relevant four visual ERP components were identified: P1, N170 and P2 from the occipital electrode cluster and the P3 component from the cluster of central electrodes, as marked in Figure 3. Each component was parameterized with four measures: peak amplitude, mean amplitude, peak latency and fractional 50% peak latency, all within a certain time interval. The P1 amplitudes and latencies were extracted within 50–150 ms interval, the N170 within 100–200 ms interval, the P2 within 200–325 ms interval and the P3 within 250–500 ms interval. Time intervals were chosen based on the previous research and the collapsed localizers approach [52], that is the zero-crossings of age group and stimulus averaged ERP waveform. Further mathematical details of the time-independent feature construction can be seen in Appendix B. As in the first dataset, the number of features was reduced from the original 16 features to 8 features, as shown in Appendix D. Based on the mutual information, both latency measures were found to be superior to amplitude measures. Nevertheless, instead of just highly correlated latency measures (see Appendix D for visual comparison), we decided to keep the best amplitude and the best latency measure, which were consistently the peak amplitude and the fractional 50% peak latency. All in all, the second dataset was comprised of 10214 trials (split between 70 participants) × 8 features. The features were the peak amplitudes and the fractional 50% peak latencies at 4 electrode clusters (occipital, parietal, central and frontal). Figure 4 shows the chosen trial-averaged features, grouped by age and stimulus type. The mean amplitudes (MA) and the peak latencies (PL), which were not included in classification tasks, are shown in Appendix A.

### 2.7. Classification Task

The task was a single-target classification with the stimulus type (frequent/rare) as the binary target. Among the trials, 124 (84%) trials were labeled as frequent (presentation of a white square) and 23 (16%) as rare (presentation of Einstein’s face) per participant. In the case of temporal features, classification was performed on each participant and time point independently, while in the case of time-independent statistical ERP features, time was not considered. The data were first split using a 10-fold cross-validation procedure. During each of the ten folds, the training data were then balanced for stimulus type to avoid majority class bias. The largest class (trials with frequent stimuli) was randomly undersampled to the average of both classes, while the minority class (trials with rare stimuli) was oversampled to the same value using SMOTE, Synthetic Minority Oversampling Technique [53]. That procedure allowed training the classifier on two equally sized classes, each with 66 trials. The training data were then shuffled. Finally, before training, the min-max normalization was performed on the training folds and applied on both train and test sets.

Six classical machine learning algorithms were tested on the task—three linear, namely LDA, SVC with linear kernel (SVC-Lin), and LR, three nonlinear, namely SVC with RBF kernel (SVB-RBF), KNN (k = 3) and a tree, as well as three ensemble methods: RF, AdaBoost and XGB, each set to 100 estimators and the maximum depth of 4. If not otherwise noted, the classifiers were applied with default parameter settings, either from the scikit-learn [54] or the xgboost module [55] in the case of XGB. Deep learning methods were not evaluated on this task due to the typically large number of parameters that would need to be optimized, for which insufficient training data were available.

Classifiers were evaluated based on their accuracy, precision, recall, F1 score, and the area under the receiver operating characteristic (AUROC) metrics. Further mathematical details of how the evaluation metrics were calculated can be seen in Appendix C. Although we show the results for all metrics, we focus primarily on the AUROC score due to the highly unbalanced dataset [56,57]. The classifiers were statistically evaluated based on the AUROC scores of each participant using the critical difference diagram [58]. Critical difference measure includes the Friedman test with the corresponding post hoc Wilcoxon tests for pair-wise comparison between the classifiers. Statistical results were corrected for multiple comparisons using Holm’s method [59]. Age-related statistical comparisons were carried out using ANOVA and post hoc independent t-tests. Lastly, feature importance was calculated using a permutation-based technique to assess the relative contribution of each feature to the classifiers’ performance.

## 3. Results

### 3.1. Differences between Classifiers

Figure 5 shows the performance scores (accuracy, AUROC, precision, recall, and F1) for each dataset type and classifier separately, averaged among participants. The results for the dataset with time-independent features are depicted by the filled circles around 0 ms, while the results for the dataset with temporal features span over the time axis as a line plot. Firstly, as expected, we observed a noticeable discrepancy between accuracy and AUROC metrics. The accuracy scores were much higher compared to AUROC scores. For example, the best accuracy scores for RF were 86.7% with the time-independent features and 78.4% with the temporal features, while the AUROC scores were 78.9% and 71.3%, respectively. Another discrepancy was that almost all classifiers had above-chance accuracy even before the stimulus appeared (e.g., RF had an average accuracy rating of around 65% before 0 ms). As already mentioned in the Methods, the most likely reason for such a non-intuitive result is the influence of imbalanced datasets. The accuracy score is highly affected by the imbalance between the classes, as the score of the more prominent class overshadows the score of the less represented class, in which we are usually more interested in [60]. Although undersampling and oversampling techniques mitigated class inequalities, the test set was not balanced, which also greatly affected the performance evaluations.

This interpretation is also supported by the precision and recall scores, or their F1 weighted average metric. All the classifiers had a poor precision score, meaning that there were relatively many false positives (frequent stimuli identified as rare). As an example, the RF had a precision score of 61.3% with the time-independent features and 42.6% with the temporal features. On the other hand, the recall scores were much better but also varied significantly between the classification models. Recall defines the ability of the classifiers to detect rare stimuli without many false negatives (rare stimuli identified as frequent). Maximum recall scores for RF were 67.7% with the time-independent features and 61.1% with the temporal features. In sum, recall/precision scores show that rare stimuli were rarely misclassified, while frequent stimuli were misclassified more often. We believe the misclassification of more numerous frequent trials gave false positives more weight in the accuracy score calculation and consequently reduced precision. The reason for the high number of false positives could also be greater noisy voltage fluctuations of single trials. As the AUROC score alleviates some of the drawbacks of the accuracy measure and is generally a better metric due to its higher discriminating power [56,57], it is the primary metric we focused on in this paper.

Secondly, all classifiers generally showed relatively low performance and moreover, classifiers showed much lower performance when trained on temporal features as compared to the training on time-independent statistical features. For example, Figure 5 shows that the best-performing classifiers were linear classifiers LDA, LR and SVC-Lin. The maximum LDA AUROC score on the time-independent dataset was 79.5%, while it was 73.3% on the temporal dataset. Similarly, the worst-performing single decision tree had a score of 73.6% for the time-independent dataset, while only 66.9% for the temporal dataset. The most likely reason for such difference is that the extraction of the relevant characteristics from the time series retains most of the information needed for the classifiers and additionally improves the robustness by removing the time variability in the data.

A third observation from Figure 5 is the temporal evolution of the decodable signal. The line plots show how approximately 50 ms after the stimulus presentation the classifiers were able to detect stimulus-related differences in the EEG signal above chance. This indicated that there were already relevant P1 component differences, enabling classifiers to separate the trials. However, the best time interval to infer the stimulus type was around 100–250 ms, which mainly corresponds to the P1 and N170 visual evoked potentials, based on the voltage fluctuations at the sensor level (see Figure 3). The performance peaks of the best three linear classifiers (LDA, LR and SVC-Lin) occurred at approximately 195 ms. After that, the performance of all models drops gradually until the end of the trial.

To statistically assess the performance of the classifiers, we performed critical difference diagram calculations using the Friedman test with the post-hoc analysis. In Figure 6A, the critical difference diagram is shown for the dataset with the time-independent ERP features. The critical distance value was calculated to be between 1.34 and 1.56. The single worst-performing classifier was a decision tree. Besides that, the three nonexclusive clusters emerged. The best-performing cluster included mainly linear classifiers, with LDA being ranked first. In Figure 6B, we extended the critical difference diagram to two dimensions, to show how the differences between classifiers vary over time. The colored markers on top of the gray background show how one of the four representative classifiers, LDA, RF, XGB or Tree, compared pairwise to all the rest. *No* marker at a specific time point represents *no* statistically significant difference (*p* > 0.05) on a multi-group Friedman test, while a marker at a specific time point represents *no* significant difference on the post-hoc pairwise test between the representative classifier and another model. In other words, statistically similar classifiers are grouped together by color. Although only four representative clusters are shown, they encompass most of the observed variability. The 2D results support the differences seen among the classifiers in Figure 5. It can be clearly observed that the classifiers did not significantly differ in their scores up to around 50 ms after the stimulus presentation. However, at approximately 100 ms, the four clusters emerged. The best cluster is represented by the LDA (marked in blue), which performed similarly well to LR and SVC-Lin. At the same time, these three models performed significantly better as all the rest, at least for several time points. Furthermore, we can observe that during the same time period, KNN classifier performed significantly worse than RF (purple), but not significantly worse than the decision tree (brown). Results nicely align with subplot A and suggest that more robust, high-bias, linear classifiers performed significantly better at this task, especially within the 200–500 ms time window.

Finally, the duration of the classification tasks is shown in Table 1 for each of the datasets. The training using the larger dataset with temporal features took from 28 min for the fastest decision tree, LDA, and LR to the slowest ensemble classifiers, which took up to 325 min (RF). As expected, the training times were much faster for the time-independent dataset, for which the loop over the time points was not needed. Again, the fastest were the linear classifiers and the decision tree with less than half a minute, while the ensemble classifiers took the longest, up to almost 18 min. These results clearly show that the dataset with time-independent statistical features is practically the only viable option for real-time EEG BCI applications.

### 3.2. Age-Related Differences

Our main question addressed in this paper was related to the potential aging effects in stimulus type classification. First, we grouped the AUROC scores based on age, as shown in Figure 7 for three representative classifiers: LDA, RF and KNN. Representative classifiers were chosen based on their fundamental model structure, where LDA represented the linear, RF the ensemble and KNN the non-linear best-performing classifiers. It can be appreciated from the figure that the performance of the linear classifiers was not affected by age, while on the other hand, the ensemble and the non-linear classifiers showed decreased performance in the older age group. When using the dataset with temporal features, the differences occurred only during a short time interval between approximately 55–129 ms for RF and between 98–133 ms for KNN. However, when the compressed time-independent features were used, age highly influenced the classification performance overall. In order to make a better comparison between dataset types, we extracted participants’ best AUROC scores based on temporal features and compared them with the above-mentioned scores from the time-independent dataset. The averaged maximum scores are quantitatively shown in Table 2 and visually depicted in Figure 8. After ANOVA tests confirmed group-level statistical differences (*p* < 0.01) for all classifiers, pairwise independent t-tests were performed. Again but now quantitatively, we confirmed significant age-related differences in classification performance for the dataset with time-independent features. The performance score was on average 0.828 (SD = 0.11) in the younger group and 0.765 (SD = 0.092) in the older age group when RF was used (marked by letter *c* in Table 2). Similarly, the score was 0.805 (SD = 0.118) for the young group, and 0.743 (SD = 0.098) in the older age group when KNN was used (marked by letter *e* in Table 2). Surprisingly, however, the maximum scores for the dataset with temporal features did not differ between age groups for any of the classifiers. Furthermore, we have additionally observed that all classifiers performed significantly better for older participants when the dataset with temporal features was used (see the letters a, b and d in Table 2. These results are to some extent contradictory to the observations from Figure 5, where it was clearly shown that the classification performance using the dataset with extracted statistical ERP features outperformed the performance using the dataset with temporal features. However, the important distinction is that in Figure 5, the individual maximum values were spread over several time points and averaging seemingly reduced the performance of such an approach. Another point to notice is the higher variance of the AUROC scores within the time-invariant dataset, as compared to the alternative, which can also be appreciated from the standard deviations in Table 2. On average, the standard deviations were 0.076 and 0.106 for temporal and time-independent features, respectively.

Up to now, we have explored the age-related differences based on the AUROC metric. While the performance metric alone could not provide an answer as to why the linear models were not affected by age, permutation-based feature ranking results shown in Figure 9 provided some clues. When the LDA was trained on the temporal features, it used a variety of features in its classification task and more than the other two classifiers based its linear boundary on spectral features. The RF and KNN, on the other hand, mainly relied on the occipital and central cluster amplitudes. As the amplitudes notably differed between the age groups, their usage inevitably led to performance discrepancies. For example, the importance of the occipital amplitude for the RF was the strongest during the P1 and the N170 component (50–150 ms), as well as during the P2 component (at 235 ms) for the younger group. In the older group, however, the occipital amplitude was important only during one stronger peak at around 300 ms, representing the averaged P2 peak (see Figure 3). Similarly, we observed an age-related time shift in the importance of the central amplitude, where the timings nicely corresponded to the P3 peak in each age group—the central cluster was most important at 340 ms in the young and only at 400 ms in the older age group. Furthermore, when we examined the feature importance of the time-independent features, similar distinctions were found. The most important features in RF and KNN were the N170 fractional 50% peak latency in the young and the N170 peak amplitude together with P3 fractional 50% peak latency in the older age group. All of the features are good in discriminating the stimulus type but do carry differences between the age groups as well (see Figure 4). In the LDA model, however, besides the N170 features, the P2 peak amplitude, agnostic to age differences, was ranked as one of the most important features in both age groups.

## 4. Discussion

In this study, we investigated the possible effects of aging on classification performance in discriminating rare and frequent trials in a passive visual oddball paradigm. To assess the aging effects more broadly, we used nine different classification models, as well as two types of datasets, one with statistical time-independent ERP features and another with temporal amplitude and spectral features.

Since many classification models were used in the analysis, we first compared their performances based on the AUROC score, which was most robust to imbalanced data. Generally, all commonly used machine learning algorithms have already been used in EEG classification tasks. A recent review has shown that, besides the deep learning algorithms, the support vector classifier (SVC) is the preferred supervised algorithm, performing with the highest accuracy [57,61]. LDA, KNN, and RF models were also noted as one of the best choices for the EEG signals. In our task, linear classifiers outperformed other classifiers, with LDA being ranked best. We assume this is because they form simpler linear decision boundaries that need fewer parameters to train, while more complex models, such as ensemble classifiers, overfit more easily when there are only a few training data available [57]. Such a result was expected, as the task included only a scarce number of especially target trials per participant. Additionally, linear models have high bias and thus the advantage when dealing with noisy and weak information signals, such as single-trial EEG data.

Next, we compared the performance between time-independent features and temporal features. In general, in BCI applications, processing speed is very important, so it is more efficient to use compressed time-independent features. On the other hand, temporal features can be used to study the temporal evolution of the signal and better understand feature importance [6]. Based on the results, the performance with the time-independent features seemed to be better at first glance, as we compared the average scores over the participants. This was expected since the extraction of relevant signal characteristics should preserve most of the information needed for the classifiers and additionally remove the time variability of the data. Interestingly, however, further analysis showed that the maximum AUROC scores of individual participants were comparable between datasets and were even significantly higher when the temporal features were used in the older age group. Another downside we observed with the time-independent features was a higher variance in performance. All in all, while the speed guarantees the usage of compressed time-independent features in the BCI applications, the presented results show that such feature extraction also brings some issues.

The primary issue we encountered was the choice of the time window for calculating the ERP parameters of each component, as the time interval can greatly affect the feature extraction and consequently the final results. One reason is that each of the ERP parameters prefers specific settings for optimal extraction [52]. The peak amplitude requires stretched measurement window to encompass inter-subject variability, while the mean amplitude functions better at shorter, specific time windows that only encompass the measured peak. This discrepancy can be also observed in our data. Although we followed well-accepted guidelines [52], high variability within as well as between the age groups led to very broad time windows. Consequently, the calculated mean amplitudes were all close to 0 µV (see Appendix A, even though the ERPs were prominent. Thus, in our case, the peak amplitude measure outperformed the mean amplitude, even though the mean amplitude is a rather preferred measure compared to the less noise-resistant peak amplitude. This has been also confirmed by the feature selection algorithm, which ranked the peak amplitude above the mean amplitude for all four components.

Another point to note is that the P3 was traditionally thought to be the most responsible component for the discrimination of target and non-target trials in the oddball study. While some studies still have such observations [57], others are showing that earlier VEP can also have the most discriminating power [62]. In our experiment, we observed that the trials can be discriminated above chance level as early as 50 ms after the stimulus presentation and are best classified at around 195 ms, at the time of N170, which is much earlier than the emergence of the P3 component. We believe that the reason for such mixed results lies in the type of visual oddball task at hand. When a very different visual stimulus is presented for the target than it is for the non-target stimulus, there is a high probability that the difference will already be observed in the visual evoked potentials.

Moreover, it has been shown that the famous phenomenon of non-learners might be due to external reasons and that all people can learn to use at least one form of BCI [63]. The same authors also show that the demographic parameters, handedness, vision correction and BCI experience have no significant effect on the performance of VEP-based BCIs. What about aging?

Finally, we focused on the effect of aging on classification performance. While the suggested reasons for the poorer performance of the older age group have been the longer reaction times and slower learning [46], we show that aging per se does not necessarily affect classification performance. Instead, the effects of aging depend on the choice of the classifier and, to some extent related, on the choice of features. If the most informative features for the classifier contain age-related differences, this will most likely lead to classification differences as well. In our dataset, ERP waveforms show the clear age-related amplitude and latency differences already at the sensor level. Some classifiers, such as LDA, used features, which were less affected by age and consequently the performance between the age groups did not differ. Moreover, LDA picked up important information from various features and thus made more robust estimates by reducing potential within-class differences, such as age. To conclude, by the choice of features that are effectively discriminating stimulus type (or any other class of interest) but are agnostic to the possible within-class differences such as aging, we can increase the classification reliability for all participants. This remark is progressively more important, as we more commonly merge data of all participants to obtain larger training sets for larger and more complex classifiers. Consequently, the training is not participant-specific, but rather cross-participant transfer learning is needed and assumed. In such cases, the feature importance of the used classifier should be checked to avoid potential performance loss in a certain subclass due to age-related or other subclass-specific characteristics.

There are also weaknesses of the study that needed to be taken into account. Besides the standard classifiers, EEG-specific classifiers have emerged in recent years (e.g., adaptive classifiers) to account for the noisiness and non-stationarity in the signal, as well as limited training data [6]. While the adjusted models would most likely increase the single-trial classification performance, we believe that the effects of aging, which were investigated here, would not differ substantially. Additionally, the data were preprocessed using the computationally costly ICA, discarding the signal related to eye and muscle artifacts. Again, however, as mostly the posterior middle electrodes were selected as the best features, we believe the eyes and muscle-related artifact would not induce larger changes in the final conclusions. The greater limitation of the experiment and analysis is that participants completed relatively few trials, which reduced the signal-to-noise ratio and decreased the learning capability of the classifiers, especially the more complex ones that would need more training data. The aging effects are planned to be further examined on a larger dataset, which would also enable us to examine the potential effects on the source-based features.

## Figures and Tables

**Figure 1 life-13-00391-f001:**
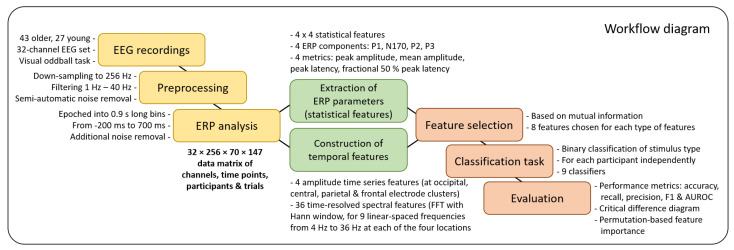
Workflow diagram.

**Figure 2 life-13-00391-f002:**
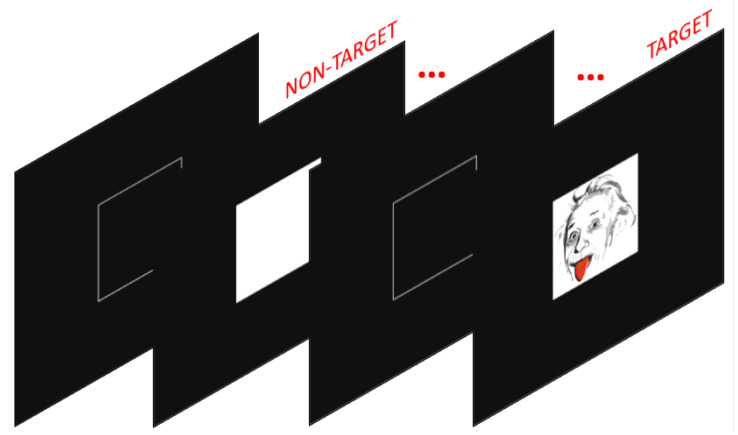
Visual oddball task.

**Figure 3 life-13-00391-f003:**
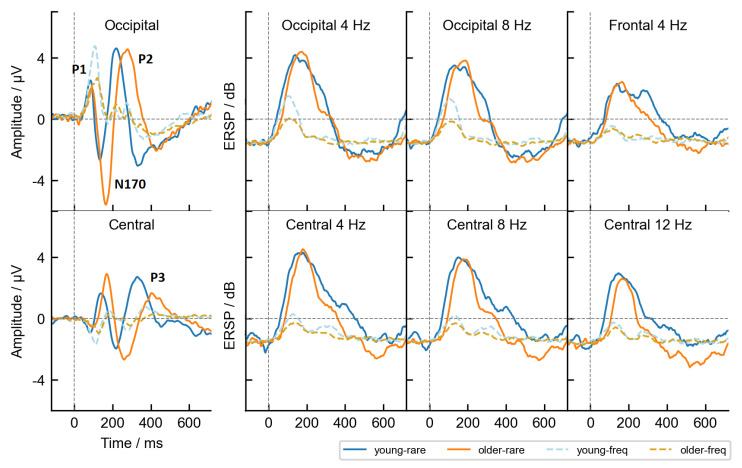
Trial-averaged temporal features grouped by age and stimulus type. The groups are distinguished by color as depicted in the legend. The first column displays amplitudes for the occipital electrode cluster (top row) and the central electrode cluster (bottom). In these subplots, the ERP components used in the second dataset are annotated in bold. The remaining subplots show changes in spectral power over time, at frequencies specified in the title of each plot. A vertical gray dashed line at 0 ms marks the onset of stimulus presentation. ERSP: Event-related spectral perturbation, shown in decibels (dB).

**Figure 4 life-13-00391-f004:**
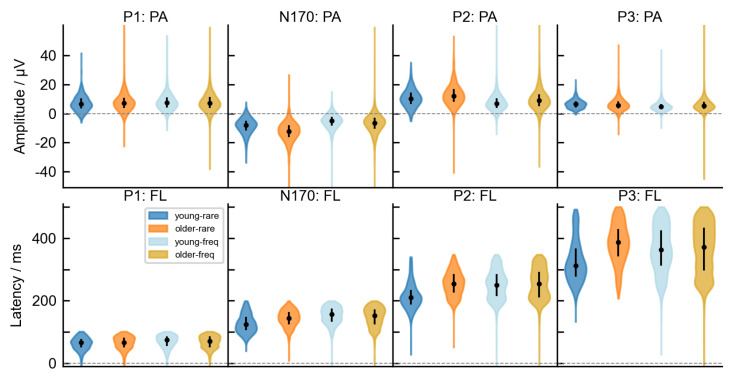
Trial-averaged time-independent features grouped by age and stimulus type. The groups are color-coded as indicated in the legend. The peak amplitudes (PA) are shown in the top row and the fractional 50% peak latencies (FL) are shown in the bottom row. The P1, N170, P2 components were calculated from the occipital electrode cluster, while the P3 component from the central electrode cluster.

**Figure 5 life-13-00391-f005:**
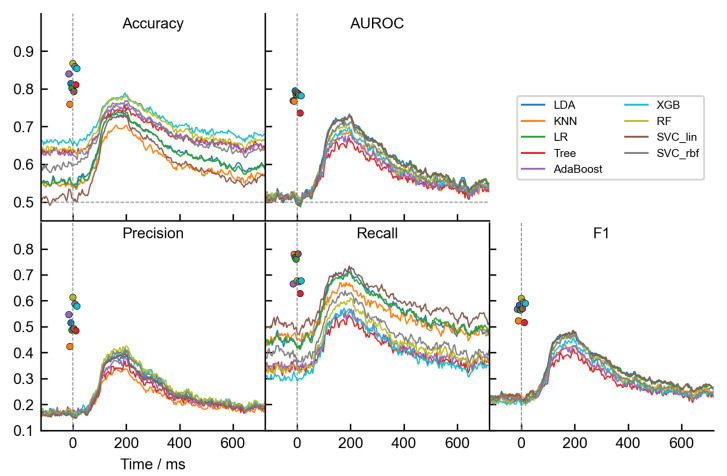
Classifiers’ performances based on five metrics: accuracy, AUROC, precision, recall and F1 score. Results for both datasets are presented. The lines show the results for the dataset with temporal features, averaged over participants. The filled circles represent the averaged results based on the time-independent features. These results, also averaged over participants, do not include time dimension and are jittered around 0 ms just for presentation purposes. Classifiers are color coded as shown in the legend. The horizontal gray dashed line at 0.5 indicates the threshold level at which the binary classifier performs at random. The vertical gray dashed line at 0 ms marks the onset of the stimulus presentation.

**Figure 6 life-13-00391-f006:**
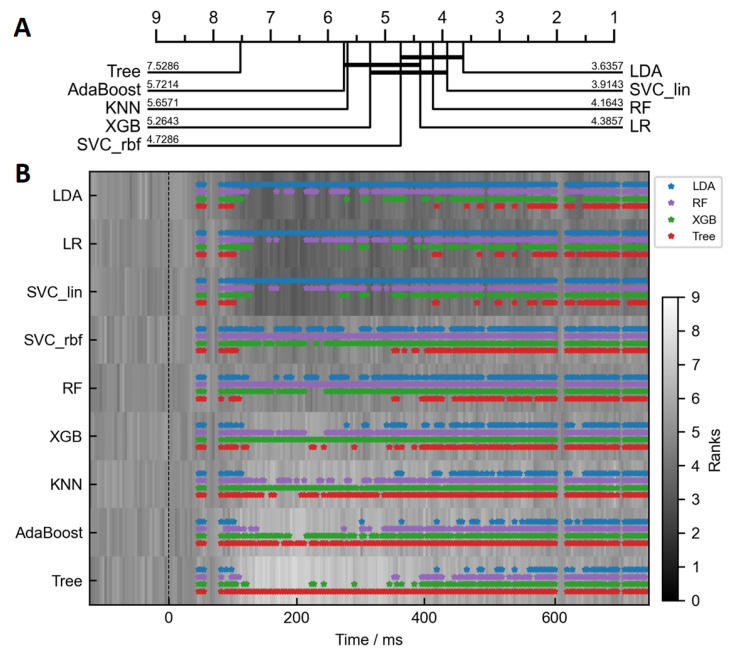
Statistical comparison of AUROC scores between classifiers using critical difference diagram. (**A**) The critical difference diagram compares the classifiers trained on time-independent features. (**B**) The comparison of the classifiers that were trained on each time point independently. The background color represents the rank of a specific classifier at a specific time point. The lower the rank (the darker the color), the better the classifier performed compared to others. If the markers on top are present, the Friedman test was statistically significant for a specific time point. The markers on top represent pairwise post hoc statistically *non-significant* differences (*p* > 0.05), meaning that at each time point, the similarly performing classifiers are grouped together. In this plot, only four representative clusters are shown, LDA (blue), RF (purple), XGB (green) and KNN (brown).

**Figure 7 life-13-00391-f007:**
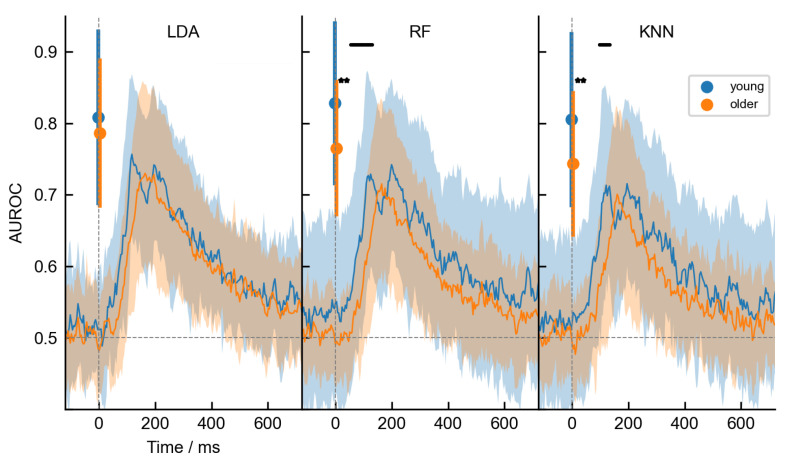
Age-related differences in AUROC scores of the three representative classifiers (LDA, RF, and KNN). The plots contain results for both datasets. The results for the training on the dataset with time-independent features are represented at 0 ms by filled circles representing the average and the error bars representing the standard deviation. Significant differences in the AUROC score between the young (blue) and older (orange) populations are shown by the two black stars (*p* < 0.01) next to the error bars. The results for the dataset with temporal features are presented as line plots. The thick line represents the average, while the shadowed area around represents the standard deviation over the trial duration. Black horizontal thick lines on the top mark the time intervals in which significant age-related differences in performance were observed. The vertical gray dashed line at 0 ms marks the onset of stimulus presentation, while the horizontal gray dashed line at 0.5 represents the chance level performance.

**Figure 8 life-13-00391-f008:**
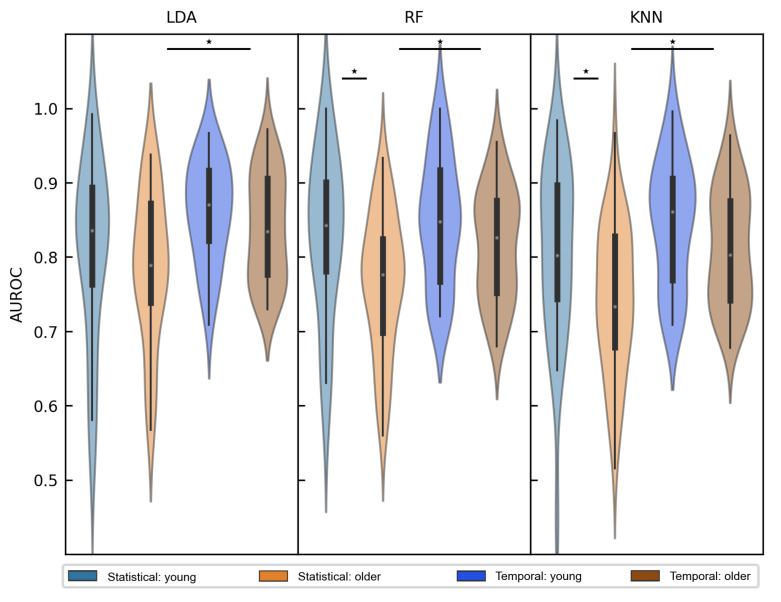
Individual subject’s best AUROC scores, grouped by age (young/older) and classification dataset type, that either includes features across time (labeled *Temporal*) or time-independent ERP parameters (labeled *Statistical*). Results are shown for the three representative classifiers (LDA: **left**, RF: middle, KNN: **right** column). Significant difference between two groups is represented by a star (*p* < 0.05).

**Figure 9 life-13-00391-f009:**
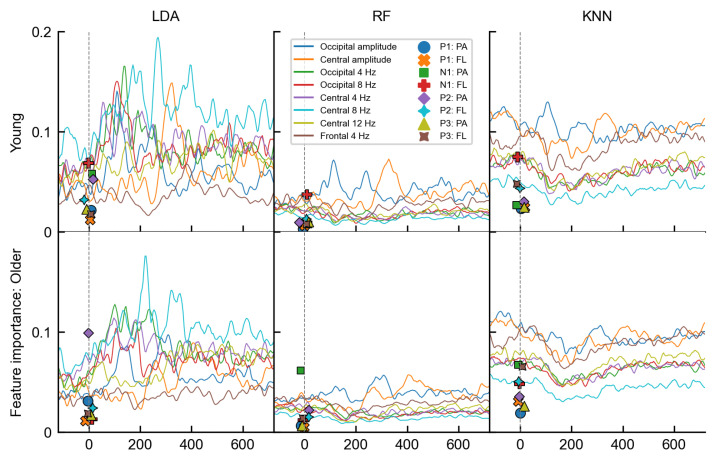
Permutation-based feature importance results. Each column shows the results of a representative classifier (LDA: **left**, RF: **middle**, KNN: **right** column). The line plots show temporal evolution of feature importances, separated for young (**top row**) and older participants (**bottom row**). The vertical gray dashed line at 0 ms marks the onset of stimulus presentation. The importances of time-independent features are depicted with various marker styles at 0 ms. Each feature is color coded as shown in the legend. Note that the N170 component is labeled as N1.

**Table 1 life-13-00391-t001:** Duration of classification tasks, measured in minutes.

	LDA	LR	SVC_lin	SVC_RBF	RF	XGB	KNN	AdaBoost	Tree
Temporal	29	29	47	63	325	110	139	313	28
Statistical	0.3	0.3	0.4	0.6	17.6	1.4	2	3.1	0.3

**Table 2 life-13-00391-t002:** Averaged individual participant’s maximum AUROC scores grouped by age and dataset type. Standard deviation is shown in parentheses. Statistically significant pairwise differences (*p* < 0.05) are marked by bolded superscript letters.

		LDA	RF	KNN
Temporal	Young	0.860 (0.068)	0.848 (0.084)	0.846 (0.083)
	Older	**0.839** ${}^{a}$ (0.072)	**0.817** ${}^{b}$ (0.074)	**0.812** ${}^{d}$ (0.077)
Statistical	Young	0.808 (0.118)	**0.828** ${}^{c}$ (0.110)	**0.805** ${}^{e}$ (0.118)
	Older	**0.786** ${}^{a}$ (0.101)	**0.765** ${}^{b,c}$ (0.092)	**0.743** ${}^{d,e}$ (0.098)

## Data Availability

Raw and preprocessed data, as well as datasets directly used in the classification tasks are located https://doi.org/10.5281/zenodo.7495536 (accessed on 30 January 2022). The code used for the analysis is available https://github.com/NinaOmejc/VEP_classification_aging.git (accessed on 30 January 2022).

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
