# Peer review of "On the Influence of Aging on Classification Performance in the Visual EEG Oddball Paradigm Using Statistical and Temporal Features"

_life, 2023, doi:10.3390/life13020391_

Round 1

Reviewer 1 Report

- There is no comparison to the related literature, and in general, the related work section need expansion.

- Although the manuscript mentions the number of features, the nature and name of these features were not mentioned. Also, it would be great if the relative importance and contribution to the classification performance is evaluated (e.g., principal component analysis). 

- It would be worthwhile to do evaluation for different subjects than the ones used in training. Also, a 10-fold cross validation invovles a small size of testing set, which may result in fluctuating performance. 

- Minor typos, e.g., train data --> trainning data 

- The performance indices need definition.

- Similar studies relating to EEG event detection can be cited so that to established the trustworthiness of the models and can provide reliabilit to baseline settings, see Gauging human visual interest using multiscale entropy analysis of EEG signals. J Ambient Intell Human Comput 12, 2435–2447 (2021). https://doi.org/10.1007/s12652-020-02381-5 and Detection of K-complexes in EEG waveform images using faster R-CNN and deep transfer learning. BMC Med Inform Decis Mak 22, 297 (2022). https://doi.org/10.1186/s12911-022-02042-x

- The table of abbreviations is missing but required by the template. 

Author Response

Dear reviewer, 

We would like to thank you for your critique and efforts, as we feel that several issues have been raised and addressed in this revision. Please, see our reply to the revision in the attachment.

Sincerely, 

Nina Omejc on behalf of all co-authors 

Reviewer 2 Report

The study analyzed the use of machine learning methods to asses the effects of age on EEG signals. The paper needs to be revised and improved according to the comments and suggestions provided below before it could be considered for publication.

Comments:

1.       Clearly state the novelty and contribution of this study in the research field.

2.       Age is one of biometric traits. Some discussion on the use of EEG signals for biometrics is needed. 10.1109/JSEN.2018.2885582, doi:10.1049/pbhe019e_ch5

3.       Present your methodology as a workflow diagram.

4.       Describe the course and details of experiment (ie., the experiment protocol) in more detail.

5.       Present your motivation (with supporting references) for the selected machine learning classifiers (like previous examples of successful use in a similar context or for similar experiments).

6.       Present the mathematical descriptions and definitions of the temporal and statistical features used in this study.

7.       Figure 4: provide the critical distance value for Figure 4a diagram.

Author Response

(The authors gave the same response as above.)

Reviewer 3 Report

In this article, the authors analyzed electroencephalograms from healthy individuals of different ages and identified that different classifiers have different performances to analyze the event-related potentials. Importantly, it is highlighted that the effect of aging depends on the classification method chosen. The article has pointed out some important questions, and the data analysis seems thorough and valid. Below are some comments to help the audience understand the results of the manuscript.

1.       In lines 189-190, the authors mentioned several evaluation metrics including accuracy, precision, recall, F1 score, and AUROC. Could the author briefly explain the calculation and meaning of these parameters extracted? This helps the readers to understand why comparing the exact values between these metrics is important, which is discussed in Section 3.1.

2.       In the abstract, the abbreviated term “EEG” should be explained when first used.

3.       In line 13, the author wrote, “Accordingly, if the model favors the features with high within-class differences, the performance will also differ.” It seems there are missing words after “high”. Careful proofreading of the manuscript is recommended.

Author Response

(The authors gave the same response as above.)

Round 2

Reviewer 2 Report

The manuscript can be accepted for publication.